# STARTrack: Learning Spatio-Temporal Representation Evolution for Target-Aware Tracking

## Abstract

Efficient modeling of spatio-temporal representations in videos is crucial for achieving accurate object tracking. Existing popular one-stream tracking frameworks typically introduce memory mechanisms or specialized modules for temporal modeling. However, due to the gradual degradation of the initial template and unstable updates of target representations, their performance often deteriorates over time. To address this issue, we propose a simple yet effective video-level tracking framework, **STARTrack**, which realizes the temporal evolution of target and context representations through an iterative token propagation mechanism. Our framework takes as input the features of the search frame along with two types of tokens that carry historical representations, and employs a visual encoder for joint modeling. This design enables target-aware perception and adaptively fuses current and historical representations. The proposed method explicitly avoids the sustained reliance on the initial template during long-term tracking, without introducing additional complex context inputs or motion modeling modules, thereby achieving faster inference. Furthermore, we develop a training strategy tailored to our framework. It enhances the semantic coherence of target representations over time via a representation consistency constraint, and for the first time explicitly incorporates occluded frames into the training process. This guides the tracker to learn context representations that are highly correlated with the spatio-temporal state of the target, thereby reducing the reliance on target appearance itself. Extensive experiments on standard benchmarks demonstrate that STARTrack achieves state-of-the-art performance, while maintaining a favorable balance between accuracy and efficiency. The code will be released.

## 1 Introduction

Visual object tracking is a fundamental yet critical task in computer vision, aiming to continuously identify and localize a specific target throughout a video sequence, given its initial position in the first frame. It plays a vital role in various real-world applications, such as video surveillance (Cheng et al., 2021; Shehzed et al., 2019), activity recognition (Aggarwal & Xia, 2014), and autonomous driving (Ettinger et al., 2021; Premachandra et al., 2020). Despite significant progress, visual tracking still faces numerous challenges, including appearance variations, distractors with similar appearances, and occlusions.

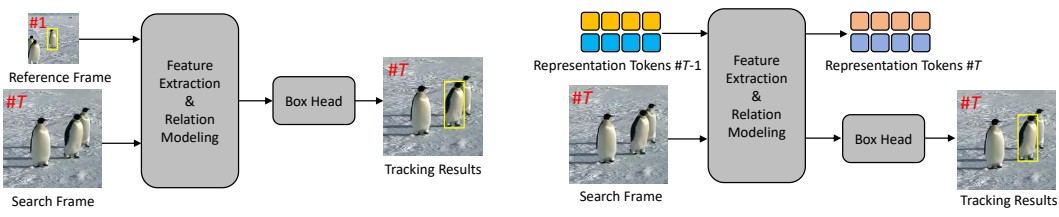

Figure 1: Comparison of different tracking methods.

Many classical visual trackers predominantly rely on an image-pair matching paradigm (Bertinetto et al., 2016; Ye et al., 2022), where the target's appearance extracted from the reference frame is matched with subsequent search frames based on appearance similarity, as illustrated in Fig. 1(a). This paradigm has achieved promising performance in standard scenarios. However, in complex environments or long-term tracking tasks, the temporal validity of the static target appearance gradually degrades, limiting adaptability to the evolution of the target. Although recent state-of-the-art trackers (He et al., 2023; Zheng et al., 2024) have introduced additional spatio-temporal cues on top of this paradigm, such as employing multiple dynamic templates or conducting cross-frame associations to alleviate representation degradation, the inherent limitations of this paradigm still restrict further performance improvements.

In addition, trackers based on the image matching paradigm typically adopt a sparse sampling strategy (Xie et al., 2024; Cai et al., 2024), since the spatial appearance of the target changes only slightly over short time intervals. Densely sampled frames may cause the model to overfit to static appearances, thereby compromising its generalization ability. However, such sparse sampling also hinders the model's capacity to capture the continuous evolution of context information that is highly relevant to the target, ultimately causing the tracker to degenerate into a simple appearance matching mechanism.

To address the aforementioned issues, we first redefine the tracking task as learning the evolution of target and context representations throughout video sequences, where the target representation refers to discriminative and robust spatio-temporal appearance cues of the target, and the context representation captures the spatio-temporal relationships between the target and other instances in the scene. Based on this perspective, we propose **STARTrack**, which departs from the conventional assumption of matching the target's fixed appearance with the search frame based on visual similarity. Instead, it performs frame-by-frame target localization and state prediction based on spatio-temporal target representations and context representations, as illustrated in Fig. 1(b). Specifically, we design two distinct types of tokens to separately store target and context representations. A visual encoder is introduced to perform deep interaction between the current search frame and both types of tokens that carry historical information. The updated tokens are then propagated to the next frame, where the process is repeated, enabling temporal update and iterative propagation of the representation tokens. This mechanism effectively models the dynamic evolution of both the target and its spatio-temporal relationships.

Unlike traditional image-matching paradigms, the design of STARTrack enables it to model the spatio-temporal evolution of both target and context representations from densely sampled video sequences. To enhance the temporal consistency of representations, we propose the Frame-wise Information Gain Principle (FIGP) and introduce a Dual-Stream Causal Representation Fusion (DCRF), which ensures that the tracker predicts the current frame solely based on temporal information from past frames, while simultaneously extracting spatio-temporally valid target and context representations from the current frame for token fusion. Notably, the proposed tracking framework introduces occluded frames into the training process for the first time, effectively guiding the model to focus on spatio-temporal context representations independent of the target's appearance and leveraging them to achieve precise target localization.

The main contributions of our work are as follows:

- We propose STARTrack, a concise and efficient video-level visual tracking framework. Through a token-based iterative propagation mechanism, STARTrack achieves elegant spatio-temporal learning of target and context representations and performs precise frame-by-frame state estimation. This design explicitly mitigates the cumulative degradation of the static target representation during long-term tracking.

- We construct a novel training strategy by introducing the Frame-wise Information Gain Principle and designing a Dual-Stream Causal Representation Fusion Mechanism. Notably, we are the first to incorporate occluded frames into the training process, facilitating high-quality propagation of both target and context representations, while ensuring their temporal consistency and effectiveness.

- STARTrack achieves state-of-the-art performance on six popular visual tracking benchmarks, including GOT-10K, TrackingNet, LaSOT, LaSOText, TNL2K, and UAV123, fully demonstrating the strong generalization and robustness of our proposed method.

## 2 RELATED WORK

### 2.1 REPRESENTATION LEARNING-BASED TRACKERS

Early methods (Bertinetto et al., 2016; Li et al., 2018; Chen et al., 2022b) commonly adopted Siamese network architectures. With the advent of Vision Transformers (Dosovitskiy et al., 2020), Ye et al. (2022) pioneered a unified framework that integrates feature extraction and relation modeling, significantly improving tracking performance. Most of the current image-level trackers (Chen et al., 2022a; Yang et al., 2023; Deng et al., 2024) further incorporate auxiliary information to enhance the modeling of the target representation, such as introducing multiple dynamic templates to perceive appearance changes (Shi et al., 2024; Kang et al., 2025), jointly modeling multiple search frames for cross-frame context association (He et al., 2023; Zheng et al., 2024; Cai et al., 2024; Gao & Wang, 2023), or generating target coordinates in an autoregressive manner to capture spatio-temporal trajectories (Wei et al., 2023; Chen et al., 2023; Bai et al., 2024). Fu et al. (2021); Cao et al. (2022); Xie et al. (2024) model multi-frame associations by introducing additional spatio-temporal modules. However, in long-term tracking or complex scenarios, the temporal validity of the fixed target appearance gradually degrades, limiting the further performance improvement of such methods.

STARTrack overcomes this limitation by introducing an iterative token propagation mechanism. It continuously extracts target and context representations from video streams, without relying on additional context inputs or specially designed auxiliary modules. Furthermore, the proposed Frame-wise Information Gain Principle ensures the spatio-temporal consistency and high-quality update of both target and context representations.

### 2.2 SAMPLE STRATEGY

Image-matching based trackers (Chen et al., 2021; Yan et al., 2021; Cui et al., 2022) typically adopt a sparse sampling strategy during training, with intervals up to 200 frames between the search frame and the reference frame. Some trackers (Zheng et al., 2024; Xie et al., 2024; Cai et al., 2024) that incorporate spatio-temporal modeling increase the average interval to 400 frames to simulate long-term variations in target appearance. However, excessively large sampling intervals often lead to drastic scene changes, degrading the model into a pure target appearance matcher and weakening its ability to model the dynamic evolution of the target. Although generative trackers (Wei et al., 2023; Bai et al., 2024; Xue et al., 2024) adopt dense sampling strategies by utilizing a sequence of historical bounding boxes to predict the target position in the current frame, this paradigm only performs cross-frame propagation at the coordinate level. As a result, it fails to explicitly capture the continuous evolution of target and context representations, thereby introducing complex training and heavy reliance on hyperparameter tuning.

## 3 METHOD

In this section, we introduce the proposed video-level tracking framework, STARTrack. The framework achieves streaming updates of target and context representations through iterative token propagation across consecutive frames.

### 3.1 MOTIVATION AND PROCESS FORMULATION

Given a set of reference frames $\{R_0, \cdots, R_t\}$ and one search frame $S_T$, the tracking procedure of a generic tracker $F$ can be expressed as $F(R_0, \cdots, R_t, S_T) \to B_T$, where B denotes the predicted box coordinates of the current search frame. Bertinetto et al. (2016); Ye et al. (2022) adopt a single reference frame, namely the target appearance in the first frame of the sequence, with the temporal index initialized as $t = 0$. On this basis, He et al. (2023); Kang et al. (2025) have introduced manually designed update strategies to select multiple reference frames from the sequence, thereby enhancing tracking performance. However, performing deep cross-correlation between multiple reference frames and the search frame requires substantial computational resources. To address this, some studies (Wei et al., 2023; Zheng et al., 2024; Xu et al., 2025) employ token storage and

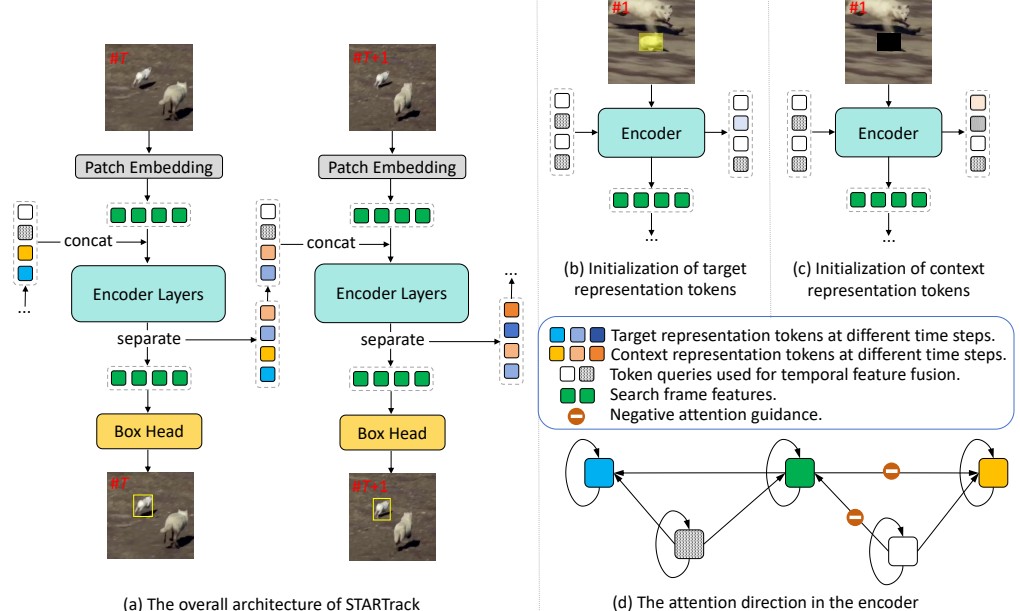

Figure 2: The tracking process of our framework.

temporal information propagation, which can be represented as $F(R_0, S_T; \{\phi_{T-1}^i\}_{i=1}^N) \to B_T$, where $\{\phi_{T-1}^i\}_{i=1}^N$ denotes spatio-temporal tokens.

However, the temporal limitation of using the reference frame $R_0$ hinders the further improvement of these otherwise strong trackers. Although the introduction of multiple reference frames alleviates this issue, their updates still rely on hand-crafted strategies. In this work, we propose a new paradigm for video tracking that leverages tokens to collect spatio-temporal information of the target and interact with the search frame, without using any temporally lagged reference frames as input. This design leads to a concise and elegant tracker while significantly improving tracking performance. Our tracking process can be formulated as follows:

$$F(S_T; \phi_{T-1}^{tgt}, \phi_{T-1}^{ctx}, \phi_q^{tgt}, \phi_q^{ctx}) \to B_T. \tag{1}$$

## 3.2 Visual Encoder

STARTrack adopts a simple pure ViT architecture to perform spatio-temporal representation aggregation and update. ViT takes an image $S_T \in \mathbb{R}^{H_s \times W_s \times 3}$ and tokens $\{\phi_{T-1}^{tgt}, \phi_{T-1}^{ctx}, \phi_q^{tgt}, \phi_q^{ctx}\}$ as input. $S_T$ is subsequently processed by a patch embedding module that splits the image into non-overlapping patches and transforms each patch into a high-dimensional embedding $f_s^0 \in \mathbb{R}^{d_m \times N_s}$ via a learnable linear projection, where $N_s = H_s W_s / p^2$, $p$ is the resolution of each image patch. The extracted search frame features $f_s^0$ are concatenated with the tokens $\{\phi_{T-1}^{tgt}, \phi_{T-1}^{ctx}, \phi_q^{tgt}, \phi_q^{ctx}\}$ and then fed into the encoder layers, where multi-head self-attention is performed to capture intra-frame dependencies. The multi-head attention in layer $j$ is computed as follows:

$$\begin{aligned}
\text{MultiHead}(\mathbf{Q}, \mathbf{K}, \mathbf{V}) &= \text{Concat}(\mathbf{H}_1, \cdots, \mathbf{H}_n)\mathbf{W}^O, \\
\mathbf{H}_i &= \text{Attention}(\mathbf{Q}\mathbf{W}_i^Q, \mathbf{K}\mathbf{W}_i^K, \mathbf{V}_G\mathbf{W}_i^V), \\
\text{Attention}(\mathbf{q}, \mathbf{k}, \mathbf{v}) &= \text{softmax}(\frac{\mathbf{q}\mathbf{k}^T}{\sqrt{d_k}} + \text{AttnMask})\mathbf{v}.
\end{aligned} \tag{2}$$

Here, $\mathbf{W}_i^Q \in \mathbb{R}^{d_m \times d_k}, \mathbf{W}_i^K \in \mathbb{R}^{d_m \times d_k}, \mathbf{W}_i^V \in \mathbb{R}^{d_m \times d_v}$ and $\mathbf{W}^O \in \mathbb{R}^{n \times d_m}$ are learnable parameters. In our case, we use $\mathbf{Q} = \mathbf{K} = \text{Concat}(f_s^j; \phi_{T-1}^{tgt,j}, \phi_{T-1}^{ctx,j}, \phi_q^{tgt,j}, \phi_q^{ctx,j}), \mathbf{V}_G = \text{Concat}(f_s^j; \phi_{T-1}^{tgt,j}, -\phi_{T-1}^{ctx,j}, \phi_q^{tgt,j}, -\phi_q^{ctx,j}), n = 12, d_m = 768,$ and $d_k = d_v = d_m/n = 64$.

The design of the attention mask follows the methods presented in Sec. 3.3.2. and Fig. 2(d). The output feature $f_s^{j+1}$ is obtained by applying a feed-forward network to the attention output, followed by residual connections and layer normalization.

## 3.3 REPRESENTATION TOKENS

We decouple the spatio-temporal representation in the video into target representations and context representations. Specifically, the target representation refers to the discriminative appearance cues of the target across space and time, while the context representation captures the spatio-temporal relationships between the target and other instances in the scene. To collect these representations from the video stream, we employ two types of token matrices: one for the target and the other for the context. As shown in Fig. 2 (a), each token matrix contains both representation tokens and token queries. The representation tokens $\{\phi_{T-1}^{tgt}, \phi_{T-1}^{ctx}\}$ aggregate all critical historical representations and are used to assist target identification in the current search frame. The token queries $\{\phi_q^{tgt}, \phi_q^{ctx}\}$ are initialized as empty and are responsible for adaptively aggregating current search frame features and historical representations from the representation tokens.

### 3.3.1 TOKEN INITIALIZATION

To maintain a unified tracking architecture, we no longer perform separate cropping on the reference frame containing the initial bounding box. Instead, we adopt the same scaling ratio and resolution as the subsequent search frames. This strategy not only simplifies the preprocessing pipeline but also encourages the tracker to attend to richer context information. As illustrated in Fig. 2(b)(c), we apply two different weighting strategies to the target frame in the reference frame for initializing the target and context representation tokens, respectively. In specific, for target token initialization, we enhance the numerical values within the target frame to emphasize its appearance features. As for context token initialization, we apply a mask to the target frame to prevent access to the target's appearance, thereby guiding the context tokens to focus on context cues related to the target's location.

### 3.3.2 DUAL-STREAM CAUSAL REPRESENTATION LEARNING

**Causal Representation Inference.** Inspired by the Transformer (Vaswani et al., 2017), we incorporate causal attention into our visual encoder, as demonstrated in Fig. 2(d). This design enables the model to leverage historical representations from multiple semantic levels to perform temporally purified attention over the search frame features. Specifically, the search frame features $f_s^j$ attend unidirectionally to the representation tokens $\{\phi_{T-1}^{tgt,j}, \phi_{T-1}^{ctx,j}\}$ that carry historical information, allowing the model to select the most semantically relevant representations to infer the target's location and appearance, while avoiding the temporal entanglement of the representations. To avoid semantic entanglement caused by direct feature fusion, we employ token queries $\{\phi_q^{tgt,j}, \phi_q^{ctx,j}\}$ to adaptively fuse the representation tokens $\{\phi_{T-1}^{tgt,j}, \phi_{T-1}^{ctx,j}\}$ and search frame features $f_s^{j+1}$, thereby constructing a robust spatio-temporal modeling mechanism characterized by selectivity, causality, and decoupling.

**Negative Attention Guidance.** Since the context representation tokens are designed to directly attend to the spatio-temporal context associated with the target, conventional attention often converges slowly and performs suboptimally under the tracking training paradigm. Following recent advances in Stable Diffusion (Nguyen et al., 2024; Guo & Du, 2025; Chen et al., 2025), we incorporate negative attention guidance into the token attention interaction process, as illustrated in Fig. 2(d). Specifically, we apply negative guidance both from the search frame features $f_s^j$ to the context representation tokens $\phi_{T-1}^{ctx,j}$ and from the context token queries $\phi_q^{ctx,j}$ to the search frame features $f_s^j$. This is achieved by flipping the sign of the corresponding parts in the attention value $v$, thereby emphasizing the target frame and enforcing a unified optimization objective in model training.

**Dual-Stream Attention.** As described in Equation 2, we control the direction of information flow by adding an attention score mask to the self-attention matrix, in order to restrict the direct interaction between $\{\phi_{T-1}^{tgt}, \phi_q^{tgt}\}$ and $\{\phi_{T-1}^{ctx}, \phi_q^{ctx}\}$. Fig 2(d) illustrates the specific attention directions: the

attention interaction between $\{\phi_{T-1}^{tgt}, \phi_q^{tgt}\}$ and $f_s^0$ in the visual encoder performs target spatio-temporal feature updating and fusion, while the attention interaction between $\{\phi_{T-1}^{ctx}, \phi_q^{ctx}\}$ and $f_s^0$ is responsible for capturing the dynamic patterns of the context features. By decoupling the inference into parallel streams, the model maintains the independence of different semantic sources and prevents mutual interference during feature aggregation, thereby generating more robust and interpretable spatio-temporal representations.

### 3.3.3 TOKEN PROPAGATION

We denote the final output of the visual encoder as $\text{Concat}(f_s^F; \phi_{T-1}^{tgt,F}, \phi_{T-1}^{ctx,F}, \phi_q^{tgt,F}, \phi_q^{ctx,F})$, and separate the search frame features from the tokens. Compared with directly using the representation tokens $\{\phi_{T-1}^{tgt,F}, \phi_{T-1}^{ctx,F}\}$, the query tokens $\{\phi_q^{tgt,F}, \phi_q^{ctx,F}\}$ updated through dual-stream causal fusion can more effectively capture the target representation and contextual variations at the current time step, while having already been adaptively integrated with historical representations. This provides more precise guidance for target localization in subsequent frames. Therefore, we discard the representation tokens output by the visual encoder, and instead adopt the fused query tokens as the new representation tokens, which are propagated together with the search frame of the next time step for target localization. This process can be formally expressed as:

$$F(S_{T+1}; \phi_T^{tgt}, \phi_T^{ctx}, \phi_q^{tgt}, \phi_q^{ctx}) \rightarrow B_{T+1}, \tag{3}$$

where $\phi_T^{tgt} = \phi_q^{tgt,F}, \phi_T^{ctx} = \phi_q^{ctx,F}$. In this way, we achieve continuous collection of target and context representations in the form of iterative token propagation along the video stream. The tracker can progressively accumulate and update the discriminative appearance features of the target as well as its interactions with the environment throughout the entire video sequence, thereby forming a dynamically evolving spatio-temporal representation that provides continuous and robust guidance for target localization across video frames.

## 4 TRAINING STRATEGY

### 4.1 DENSE FRAME SAMPLING INCLUDES OCCLUSION

STARTrack eliminates the reliance on the target's static appearance and instead models the continuous spatio-temporal evolution of the target representation and its context representation, enabling effective learning with densely sampled frame sequences. Specifically, we select one search frame every 0.3 seconds based on the video frame rate and regroup multiple ordered search frames into a new video clip for training. Furthermore, thanks to the design of context representation tokens, we are the first to introduce occluded frames into the training process, enhancing the context tokens' perception of spatial structure. This allows the model to accurately localize the target based on context cues even under occlusion—a strategy that is infeasible under conventional image pair matching paradigms that rely solely on target's appearance.

### 4.2 FRAME-WISE INFORMATION GAIN PRINCIPLE

The image-matching paradigm typically maintains a fixed initial appearance of the target as input, which helps preserve the spatio-temporal stability of the target representation to some extent. In contrast, for the propagation paradigm, a critical challenge lies in ensuring the high-quality update and spatio-temporal consistency of both target and context representations throughout the propagation process.

Our central insight is that as long as the representation tokens are updated with sufficient quality at each frame, it is possible to obtain a stable and reliable spatio-temporal representation across the sequence. But how can we evaluate the quality of token updates? Inspired by the residual mechanism (He et al., 2016), we propose the Frame-wise Information Gain Principle: for a given search frame $S_{T+1}$, representation tokens $\{\phi_T^{tgt}, \phi_T^{ctx}\}$ are expected to capture richer spatio-temporal information than $\{\phi_{T-1}^{tgt}, \phi_{T-1}^{ctx}\}$, thus enabling more accurate predictions. Following this principle, we compare predictions by interacting the same search frame with representation tokens from different time

steps. This strategy effectively mitigates the problem of representation collapse in the propagation paradigm and facilitates stable and high-quality evolution of spatio-temporal representations over time.

### 4.3 LOSS FUNCTION

We adopt commonly used losses in the prediction head network, including focal loss (Law & Deng, 2018), GIoU loss (Rezatofighi et al., 2019) and $L_1$ loss. For target bounding box prediction in the current frame, we denote $cur$ as the prediction result obtained using representation tokens $\{\phi_T^{tgt}, \phi_T^{ctx}\}$, and $prev$ as the corresponding result from $\{\phi_{T-1}^{tgt}, \phi_{T-1}^{ctx}\}$. Compared to conventional loss designs (Ye et al., 2022), we incorporate a temporal refinement term into each loss component, and the overall loss with the Frame-wise Information Gain Principle can be computed as:

$$L = \sum_{i \in \{iou, cls, 1\}} \lambda_{L_i} [L_i^{cur} + \text{ReLU}(L_i^{cur} - L_i^{prev})], \tag{4}$$

we set $\lambda_{giou} = 2$, $\lambda_{cls} = 1$ and $\lambda_{L_1} = 5$.

## 5 EXPERIMENTS

### 5.1 IMPLEMENTATION DETAILS

**Model Configuration.** We present two variants of STARTrack with different input resolutions: STARTrack-256 (search region size: 256×256) and STARTrack-384 (search region size: 384×384). Additionally, we used DINOv2 as the backbone network to construct more powerful STARTrack-D256 and STARTrack-D384. For details on the training strategy, please refer to § A.

**Inference.** During the inference, the two types of representation tokens are updated and propagated frame by frame without any additional operations. We conduct comparative experiments in terms of model parameters, FLOPs, and inference speed, as shown in Tab. 1. Since STARTrack requires neither additional reference frame inputs nor complex module designs, it achieves an inference speed of 84 FPS at a resolution of 384, which is nearly 2 times faster than AQATrack-384. It is worth noting that STARTrack-D, which uses the same backbone as SPMTrack, has fewer parameters and lower computational cost, which results in higher inference speed.

Table 1: Comparison of model parameters, FLOPs, and inference speed, with model resolution unified to 384×384.

| Method | Type | Params | FLOPs | Speed | Device |
|---|---|---|---|---|---|
| SeqTrack | ViT-B | 89 M | 148 G | $11 fps$ | 2080Ti |
| ODTrack | ViT-B | 92M | 73G | $32 fps$ | 2080Ti |
| AQATrack | HiViT-B | 72M | 58G | $44 fps$ | v100 |
| SPMTrack | DINOv2-B | 115M | 86G | $26 fps$ | 2080Ti |
| STARTrack | ViT-B | 86M | 36G | $84 fps$ | 2080Ti |
| STARTrack-D | DINOv2-B | 92M | 48G | $70 fps$ | 2080Ti |

### 5.2 COMPARISON WITH STATE-OF-THE-ART TRACKERS

**LaSOT.** LaSOT (Fan et al., 2019) is a large-scale long-term tracking benchmark with high-resolution videos and rich attribute annotations, where the test set contains 280 diverse and challenging sequences. As demonstrated in Tab. 2, STARTrack-D256 and STARTrack-D384 achieve excellent performance compared to trackers with the same resolution, with AUC scores of 74.4% and 76.0%, respectively. For more detailed analysis of the attributes on LaSOT, please refer to § B .

**LaSOT$_{ext}$.** LaSOT$_{ext}$ (Fan et al., 2021) extends LaSOT with 150 videos featuring similar distractors and frequent occlusions, challenging the spatio-temporal modeling and discrimination of trackers. As reported in Tab. 2, our method performs better than most previous state-of-the-art trackers. For example, STARTrack-256 gets an AUC of 52.1%, $P_{Norm}$ score of 63.5%, and P score of 59.3%, outperforming the AQATrack-256 by 0.9%, 1.3%, and 0.4%, respectively. These results demonstrate the stability of our representation tokens' spatio-temporal propagation and their effectiveness in capturing densely inter-frame variations.

**GOT-10K.** The test set of GOT-10K (Huang et al., 2019) contains 180 videos with completely unseen object categories, making it a strict evaluation of generalization ability. We follow the one-shot

| Method | Source | LaSOT | | | LaSOT$_{ext}$ | | | GOT-10K* | | | TrackingNet | | |
|---|---|---|---|---|---|---|---|---|---|---|---|---|---|
| | | AUC | P$_{norm}$ | P | AUC | P$_{norm}$ | P | AO | SR$_{0.5}$ | SR$_{0.75}$ | AUC | P$_{norm}$ | P |
| STARTrack-D256 | Ours | **74.4** | **83.7** | **80.9** | **52.5** | **63.8** | **59.6** | **78.0** | **87.5** | **74.4** | **85.6** | **90.6** | **85.5** |
| STARTrack-256 | Ours | 73.8 | 83.5 | 80.4 | 52.1 | 63.5 | 59.3 | 77.8 | 87.3 | 74.1 | 85.3 | 90.2 | 85.1 |
| ARPTrack-256(Liang et al., 2025) | CVPR25 | 72.6 | 81.4 | 78.5 | 52.0 | 62.9 | 58.7 | 77.7 | 87.3 | 74.3 | 85.5 | 90.0 | 85.3 |
| ARTrackV2-256(Bai et al., 2024) | CVPR24 | 71.6 | 80.2 | 77.2 | 50.8 | 61.9 | 57.7 | 75.9 | 85.4 | 72.7 | 84.9 | 89.3 | 84.5 |
| AQATrack-256(Xie et al., 2024) | CVPR24 | 71.4 | 81.9 | 78.6 | 51.2 | 62.2 | 58.9 | 73.8 | 83.2 | 72.1 | 83.8 | 88.6 | 83.1 |
| ROMTrack-256(Cai et al., 2023) | ICCV23 | 69.3 | 78.8 | 75.6 | 48.9 | 59.3 | 55.0 | 72.9 | 82.9 | 70.2 | 83.6 | 88.4 | 82.7 |
| ARTrack-256(Wei et al., 2023) | CVPR23 | 70.4 | 79.5 | 76.6 | 46.4 | 56.5 | 52.3 | 73.5 | 82.2 | 70.9 | 84.2 | 88.7 | 83.5 |
| OSTrack-256(Ye et al., 2022) | ECCV22 | 69.1 | 78.7 | 75.2 | 47.4 | 57.3 | 53.3 | 71.0 | 80.4 | 68.2 | 83.1 | 87.8 | 82.0 |
| MixFormer-22k(Cui et al., 2022) | CVPR22 | 69.2 | 78.7 | 74.7 | - | - | - | 70.7 | 80.0 | 67.8 | 83.1 | 88.1 | 81.6 |
| STARK(Yan et al., 2021) | ICCV21 | 67.1 | 77.0 | - | - | - | - | 68.8 | 78.1 | 64.1 | 81.3 | 86.1 | - |
| TransT (Chen et al., 2021) | CVPR21 | 64.9 | 73.8 | 69.0 | - | - | - | 67.1 | 76.8 | 60.9 | 81.4 | 86.7 | 80.3 |
| *Some Trackers with Higher Resolution* | | | | | | | | | | | | | |
| OSTrack-384(Ye et al., 2022) | ECCV22 | 71.1 | 81.1 | 77.6 | 50.5 | 61.3 | 57.6 | 73.7 | 83.2 | 70.8 | 83.9 | 88.5 | 83.2 |
| SeqTrack-B384(Chen et al., 2023) | CVPR23 | 71.5 | 81.1 | 77.8 | 50.5 | 61.6 | 57.5 | 74.5 | 84.3 | 71.4 | 83.9 | 88.8 | 83.6 |
| ROMTrack-384(Cai et al., 2023) | ICCV23 | 71.4 | 81.4 | 78.2 | 51.3 | 62.4 | 58.6 | 74.2 | 84.3 | 72.4 | 84.1 | 89.0 | 83.7 |
| ARTrack-384(Wei et al., 2023) | CVPR23 | 72.6 | 81.7 | 79.1 | 51.9 | 62.0 | 58.5 | 75.5 | 84.3 | 74.3 | 85.1 | 89.1 | 84.8 |
| ODTrack(Zheng et al., 2024) | AAAI24 | 73.2 | 83.2 | 80.6 | 52.4 | 63.9 | 60.1 | 77.0 | 87.9 | 75.1 | 85.1 | 90.1 | 84.9 |
| ARTrackV2-384(Bai et al., 2024) | CVPR24 | 73.0 | 82.0 | 79.6 | 52.9 | 63.4 | 59.1 | 77.5 | 86.0 | 75.5 | 85.7 | 89.8 | 85.5 |
| SPMTrack-384(Cai et al., 2025) | CVPR25 | 74.9 | 84.0 | 81.7 | - | - | - | 76.5 | 85.9 | 76.3 | 86.1 | 90.2 | 85.6 |
| STARTrack-384 | Ours | 75.2 | 84.7 | 82.3 | 53.2 | 64.7 | 61.0 | 78.5 | **87.9** | 76.5 | 86.4 | 90.8 | 86.2 |
| STARTrack-D384 | Ours | **76.0** | **85.5** | **82.8** | **53.6** | **65.2** | **61.3** | **79.0** | **88.1** | **77.1** | **86.5** | **91.0** | **86.3** |

Table 2: Comparison with state-of-the-art trackers on four popular benchmarks: LaSOT, LaSOT$_{ext}$, GOT-10K, and TrackingNet. Where * denotes for trackers only trained on GOT-10K. Best in bold, second best underlined.

protocol and evaluate our tracking results online. As shown in Tab. 2, STARTrack-D256 outperforms all previous trackers (78.0% AO), while STARTrack-D384 also achieves competitive performance (79.0% AO). These results further validate the strong generalization capability of our method.

**TrackingNet.** TrackingNet (Muller et al., 2018) is a large-scale short-term dataset, with a test split of 511 sequences. As reported in Tab. 2, compared to most other tracking algorithms, our STARTrack-384 achieves a new state-of-the-art result. For example, compared with the latest SPMTrack-384, our simple method achieves 0.3%, 0.6%, and 0.6% gains in terms of AUC, P$_{Norm}$ and P score, respectively. These results show that our target representation tokens can capture critical appearance cues of the target, thereby enabling more accurate tracking.

**UAV123& TNL2K.** UAV123 (Benchmark, 2016) is a challenging aerial tracking benchmark consisting of 123 UAV-captured sequences, while TNL2K (Wang et al., 2021) is a large-scale dataset consisting multi-source data, including RGB-T, cartoons, and more. Results are reported in § C.

## 5.3 ABLATION STUDIES

### 5.3.1 DIFFERENT COMBINATIONS OF TOKENS

We evaluate different combinations of representation tokens for iterative propagation, as shown in Tab. 3(a). Both the context-only setting (a) (#2) and the target-only setting (#3) lead to at least a 1.4% drop in AUC score, indicating that both types of tokens play critical roles in frame-by-frame target discrimination. Notably, the performance of setting (#3) slightly surpasses that of OSTrack-256, suggesting that the target tokens in our method capture more discriminative target representations from the video compared to a fixed initial target appearance.

| (a) Study on representation tokens | | | | (b) Study on sample mode | | | | | (c) Study on training strategy | | | |
|---|---|---|---|---|---|---|---|---|---|---|---|---|
| Token Combination | AUC | P$_{norm}$ | P | Density | Ordering | AUC | P$_{norm}$ | P | Strategies | AUC | P$_{norm}$ | P |
| All | **73.8** | **83.5** | **80.4** | dense | order | **73.8** | **83.5** | **80.4** | All | **73.8** | **83.5** | **80.4** |
| w/o target tokens | 72.4 | 81.9 | 78.5 | dense | casual | 73.2 | 82.9 | 79.5 | w/o occlusion | 71.6 | 80.7 | 78.9 |
| w/o context tokens | 69.5 | 79.2 | 76.0 | sparse | order | 72.6 | 82.0 | 78.9 | w/o FIGP | 44.7 | 52.5 | 43.1 |
| w/o token queries | 71.1 | 81.5 | 77.3 | sparse | casual | 72.5 | 82.3 | 78.5 | FIGP→MIGP | 73.5 | 83.4 | 80.1 |

Table 3: Ablation studies on the LaSOT benchmark.

### 5.3.2 ATTENTION VARIANTS

As illustrated in Fig. 3, we provide three attention variants to demonstrate the critical role of dual-stream causal attention with Negative Attention Guidance (NAG) in ensuring high-quality updating and propagation of representation tokens. Compared with the attention mechanism adopted in STARTrack-384 shown in Fig. 3(a), variant (b) employs standard attention instead of causal attention, directly fusing the historical and current representations through representation tokens and iteratively propagating the updated tokens, rather than relying on a query-based mechanism. Although this design omits the use of token queries and their interactions, it introduces temporal entanglement and semantic loss in target representations, leading to performance drops across all metrics (see Tab. 3(a)(#4)). Variant (c) depicts the attention mechanism without NAG, where the inconsistency between the optimization objectives of target and context representation tokens results in training difficulties and a significant performance degradation.

To further verify the effectiveness of dual-stream attention, we construct cross-connections between the two causal attention streams. Specifically, the context token query $\phi_q^{ctx}$ is allowed to attend to the target representation tokens $\phi_T^{tgt}$ under the guidance of NAG, while the target token query $\phi_q^{tgt}$ is also enabled to attend to the context representation tokens $\phi_T^{ctx}$. This hybrid form of attention breaks the independence constraint inherent in the dual-stream design, allowing interference from context signals to seep into the target stream, thereby weakening the temporal purification effect of causal attention.

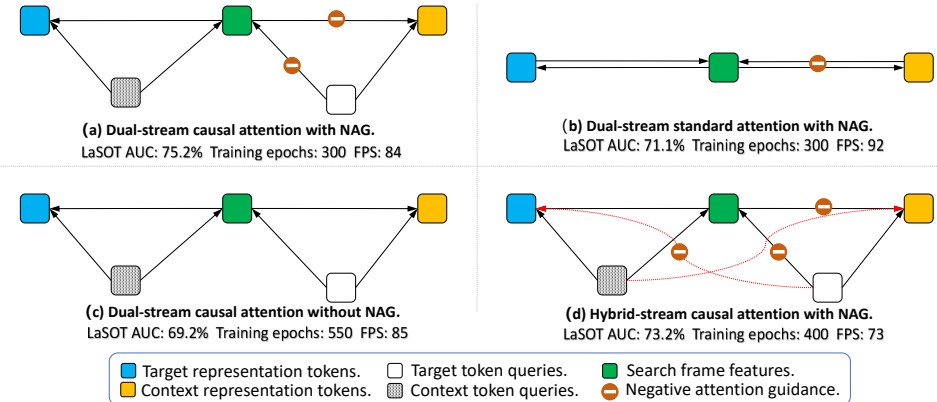

**(a) Dual-stream causal attention with NAG.**
LaSOT AUC: 75.2%  Training epochs: 300  FPS: 84

**(b) Dual-stream standard attention with NAG.**
LaSOT AUC: 71.1%  Training epochs: 300  FPS: 92

**(c) Dual-stream causal attention without NAG.**
LaSOT AUC: 69.2%  Training epochs: 550  FPS: 85

**(d) Hybrid-stream causal attention with NAG.**
LaSOT AUC: 73.2%  Training epochs: 400  FPS: 73

| | Target representation tokens. | | Target token queries. | | Search frame features. |
| Context representation tokens. | | Context token queries. | | Negative attention guidance. |

Figure 3: Attention variants explored in our ablation studies. The dual-stream setting decouples target and context representations, while the hybrid-stream merges them into a unified flow. For the sake of clarity and visual simplicity, we omit the arrows denoting self-attention in the figure.

### 5.3.3 SAMPLE MODE

As shown in Tab. 3(b), we ablate the impact of sample mode on the tracking performance. Dense sequential sampling achieves the best performance. In contrast, dense random sampling weakens the tracker's temporal awareness, resulting in a 0.6% drop in AUC score. The sparse sampling scheme further degrades STARTrack into learning only discriminative target appearances, weakening the modeling of continuous target and context representations, and thus achieves an AUC score of only around 72.5%.

### 5.3.4 TRAINING PARADIGM

The Frame-wise Information Gain Principle plays a vital role in maintaining the spatio-temporal consistency of representations. As shown in Tab. 3(c), models trained without this principle fail to preserve robust representation updates, resulting in a significant 29.1% drop in AUC score. We further extend this principle to a multi-frame version, assuming that tokens aggregated from the previous T + ΔT frames should yield more accurate predictions than those from only the previous T frames. However, this modification leads to a slight performance drop, which we attribute to unstable updates of tokens from intermediate frames. Notably, if occluded frames are excluded

during training, the AUC score drops to 71.6%, highlighting the importance of occlusion-aware training for STARTrack's context-aware capability.

### 5.3.5 INITIALIZATION OF TARGET REPRESENTATION TOKENS

Since the target location in the first frame $R_0$ of a sequence is usually provided in the form of a bounding box, an important question arises: how to indicate the target region in the reference frame without introducing additional tokens? This directly determines whether the target representation tokens can effectively focus on the target region and extract reliable initial representations. To investigate this, we conduct comparative experiments on three benchmark

Table 4: Comparison of different target enhancement methods on the reference frame for target representation token initialization.

| Target Enhancement | LaSOT | UAV123 | TNL2K |
|---|---|---|---|
| N/A | 25.2 | 27.3 | 19.6 |
| Constant | 73.8 | 71.9 | 60.3 |
| 2D Gaussian | 73.5 | 72.2 | 59.9 |
| Segment & Constant | 74.1 | 71.8 | 60.0 |

datasets by initializing target representation tokens with reference frames processed using different target enhancement strategies, as shown in Tab. 4. (#1) shows that without any numerical enhancement of the target region, the target representation tokens can hardly perceive the initial state of the target. We further compare three different enhancement strategies: constant enhancement, Gaussian edge attenuation, and bounding-box-guided target segmentation. All three approaches enable the model to perceive the target region, leading to comparable performance across the three ($\pm$ 0.4). For the sake of framework simplicity and efficient reference frame preprocessing, we adopt constant enhancement as the default strategy.

## 6 VISUALIZATION AND ANALYSIS

To intuitively demonstrate the effectiveness of our proposed method, we visualize the attention distributions of STARTrack. Fig. 4 presents challenging scenarios involving motion blur and occlusion. Notably, both scenes contain numerous similar-looking objects. Although our target representation tokens occasionally attend to incorrect instances in such cases, the context representation tokens can accurately determine the target's location by leveraging dense spatio-temporal correlations.

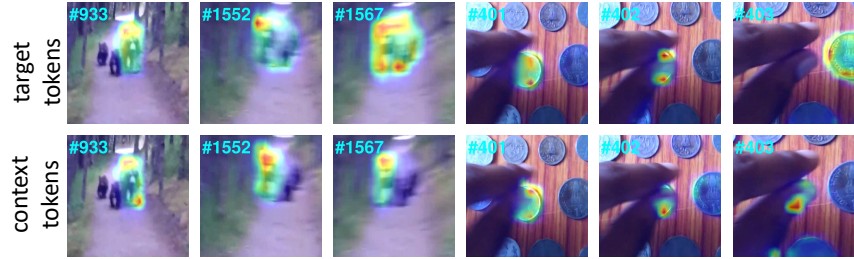

Figure 4: Visualization of the respective attention maps from target and context representation tokens to the search region in complex scenarios.

## 7 CONCLUSION

We proposed STARTrack, a concise and efficient video-level tracking framework. By leveraging an iterative token propagation mechanism, STARTrack continuously gathers and updates discriminative target and context representations across video streams, enabling precise perception and robust modeling. Our dual-stream causal attention with negative attention guidance ensures temporally purified and consistent representation updates, while the proposed training strategy—incorporating the Frame-wise Information Gain Principle, dense sampling, and occlusion-aware training—further enhances robustness and generalization in long-term tracking. Extensive experiments across six benchmarks demonstrate that STARTrack achieves state-of-the-art performance while maintaining favorable inference efficiency.

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

## APPENDIX

## A  TRAINING

We adopt the MAE-pretrained ViT-Base (Dosovitskiy et al., 2020) as the visual encoder for STAR-Track and DINOv2 (Oquab et al., 2023) as the visual encoder for STARTrack-D. The training datasets include GOT-10K (Huang et al., 2019), TrackingNet (Muller et al., 2018), and LaSOT (Fan et al., 2019). The target representation token matrices $\phi_T^{tgt}$ and $\phi_q^{tgt}$ are configured with a size of 4×4, whereas the context representation token matrices $\phi_T^{ctx}$ and $\phi_q^{ctx}$ are configured with a size of 4×4. We randomly select 8 sequences and sample 5 frames from each according to the dense sampling strategy, with the first frame serving as the reference frame, resulting in a batch size of 32. Given that videos in GOT-10K are sampled at 10 fps, we select one search frame every 3 frames, while in the other datasets, one search frame is selected every 10 frames to ensure consistent frame interval alignment across datasets. We sample 60k images per training epoch. After 30 epochs, we randomly apply target region masking to any two non-occluded search frames among the four. Training is conducted on four NVIDIA RTX 3090 GPUs using PyTorch 1.13.1 with Python 3.8, and we employ the AdamW (Loshchilov & Hutter, 2017) for gradient descent. The initial learning rate for the backbone is set to $4\times10^{-5}$, while the rest of the model uses a learning rate of $4\times10^{-4}$, with a weight decay of $1\times10^{-4}$. The total number of training epochs is 300, with the learning rate decayed by a factor of 10 after 240 epochs. Following the one-shot protocol, the model evaluated on GOT-10K is trained only on its training split, with the learning rate decayed after 80 epochs within a total training schedule of 100 epochs. For STARTrack-D, we applied LoRA (Hu et al., 2022) to fine-tune the backbone network.

## B  ATTRIBUTE ANALYSIS ON LASOT

The attribute-based results in Fig. 5 highlight the robustness of our model in complex scenarios, especially under occlusion and background clutter. This robustness stems from the design of our context representation tokens, which facilitate precise target localization by modeling context relationships and effectively suppressing background interference.

## C  RESULTS ON UAV123& TNL2K

UAV123 is a challenging aerial tracking benchmark consisting of 123 UAV-captured sequences, while TNL2K is a large-scale dataset consisting multi-source data, including RGB-T, cartoons, and more. As reported in Tab. 5, STARTrack-256 achieves leading performance on both benchmarks, outperforming the previous best trackers with an AUC improvement of 1.2% and 1.1%, respectively.

|  | Ocean | TransT | DiMP50 | STARK | OSTrack | SeqTrack | ARTrack | AQATrack | ARTrackV2 | Ours |
|---|---|---|---|---|---|---|---|---|---|---|
| UAV123 | 57.4 | 68.1 | 64.3 | 68.2 | 68.3 | 68.6 | 67.7 | 70.7 | 69.9 | **71.9** |
| TNL2K | 38.4 | 50.7 | 44.7 | - | 55.9 | 56.4 | 57.5 | 57.8 | 59.2 | **60.3** |

Table 5: Comparison with state-of-the-art methods on UAV123 and TNL2K benchmarks in AUC score.

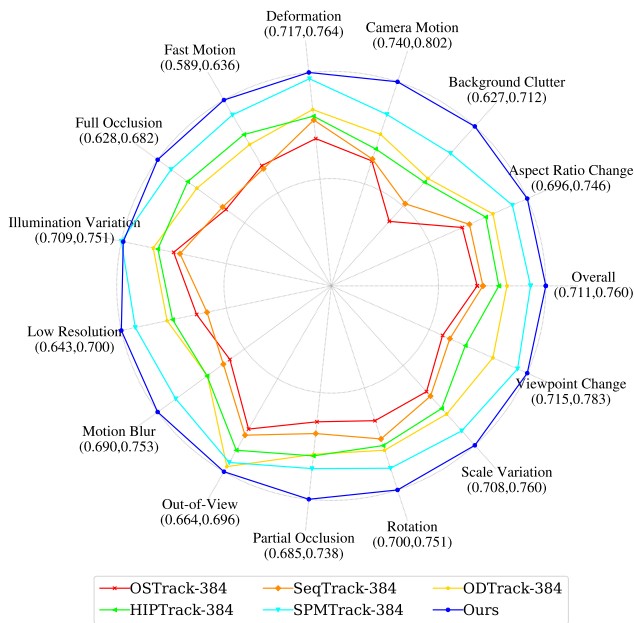

Figure 5: AUC performance comparisons across 15 LaSOT attributes. Best viewed in color.

# D MORE ABLATION STUDIES

## D.1 SAMPLE LENGTH

We conduct ablation studies on different sampling lengths and report the results on LaSOT and GOT-10k (see Fig.6), with the batch size fixed to 40. When the sampling length is set to 1, STARTrack degenerates into the conventional matching paradigm. As the sampling length increases, the performance on both datasets steadily improves, which validates that our proposed representation token fusion and propagation mechanism can effectively exploit temporal information to enhance prediction accuracy. It is worth noting that when the sampling length exceeds 20, the tracking performance degrades, which we attribute to the scarcity of very long sequence samples in the datasets. Moreover, the relatively small performance fluctuation on GOT-10k across different sampling lengths further highlights the importance of temporal representations for long-term tracking.

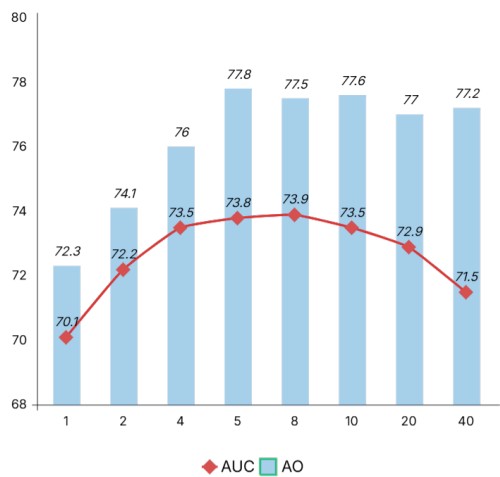

Figure 6: The ablation study on sampling length conducted on LaSOT and GOT-10k, the batch size is set to 40.

### D.2 SIZE OF THE REPRESENTATION TOKEN MATRICES

The design of the representation token matrices and token query matrices directly determines the storage and updating capacity of spatio-temporal representations. In this experiment, we keep the sizes of the two types of representation tokens consistent with their corresponding token queries, i.e., $|\phi_T^{tgt}| = |\phi_q^{tgt}|, |\phi_T^{ctx}| = |\phi_q^{ctx}|$. Fig. 7 shows the performance of STARTrack-384 on LaSOT$_{ext}$ with different target token matrices and context token matrices sizes. We can observe that, when the context representation token size is fixed, enlarging the target representation token matrix allows the tracker to capture richer spatio-temporal representations of the target, thereby improving its discriminative capability. On this basis, the inclusion of context representations further boosts tracking performance, indicating that non-target representations also play a critical role in tracking, as the spatio-temporal relations between the target and other instances in the scene are important factors. It is worth noting that when the sizes of both token matrices increase from $4 \times 4$ to $5 \times 5$, the performance gain becomes marginal. We speculate that this is because the search frames are cropped and resized based on the target bounding box, leading to relatively fixed target and context representations.

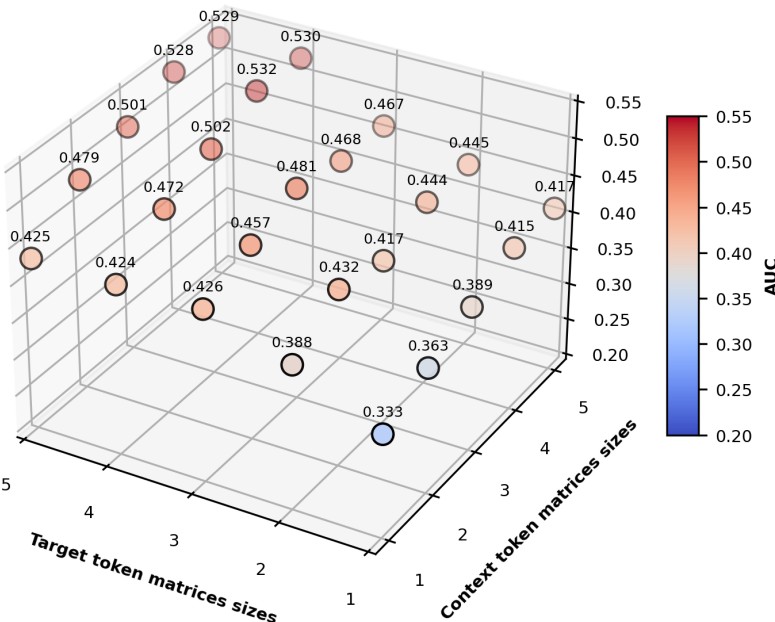

Figure 7: Comparison of the impact of representation token matrix size on STARTrack-384 performance on LaSOT$_{ext}$. The x-axis denotes the side length of the target token matrices, while the y-axis denotes the side length of the context token matrices.

## E  MORE VISUALIZATION

To intuitively demonstrate the effectiveness of our proposed method, we visualize the tracking results comparsion of STARTrack and other previous SOTA trackers. As shown in Fig. 8, compared to four state-of-the-art trackers, our approach effectively distinguishes similar objects in the scene and maintains temporal consistency of target representations, thereby enabling more accurate instance tracking.

## F  LIMITATION

Although STARTrack achieves excellent performance on various benchmark datasets, its tracking capability is still limited in extremely complex scenes. For example, in cases of long-term occlusion or when the target exits and re-enters the field of view after some time, the tracker fails to capture

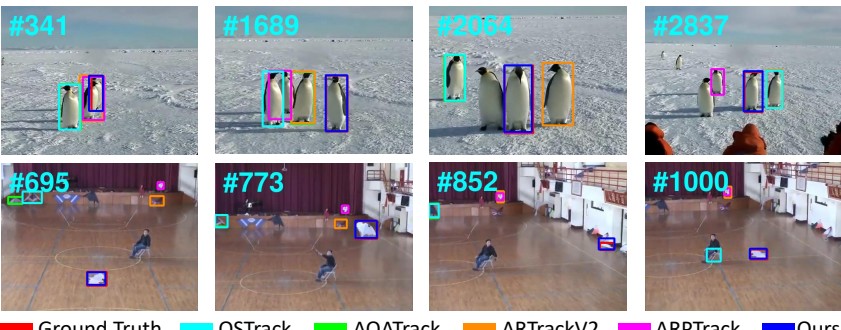

Figure 8: Comparison tracking results with other four SOTA trackers on LaSOT benchmark.

the target's appearance and motion trajectory. As a result, continuous modeling of the background region introduces slight noise into the target representation token, which reduces its representation quality, leading to tracking drift or mis-tracking of similar objects. A feasible solution is to introduce an additional mechanism for detecting target disappearance. Furthermore, due to GPU memory limitations, STARTrack trains with a batch size of 8 sequences, each containing 5 consecutive frames, which somewhat restricts the ability of representation tokens to propagate over long periods. With richer sampling sequences, better hardware configurations, and longer sampling sequences, STAR-Track's long-term tracking potential can be more effectively unlocked.

