# OpenReview forum: "STARTrack:Learning Spatio-Temporal Representation Evolution for Target-Aware Tracking"
_ICLR.cc/2026/Conference — Submitted to ICLR 2026_

### Official Review · Reviewer_cD5U · 2025-10-14

**Soundness:** 3
**Presentation:** 3
**Contribution:** 3
**Rating:** 6
**Confidence:** 5

**Summary:**

STARTRACK is a novel video-level object tracking framework that redefines visual tracking as a spatio-temporal representation evolution problem. Unlike traditional trackers that rely on static template matching or sparse temporal updates, STARTRACK introduces an iterative token propagation mechanism to dynamically model the evolution of both target and context representations across video frames.

Key components include:

Two types of tokens: target tokens and context tokens, which store and propagate spatio-temporal representations.

A dual-stream causal attention mechanism with negative attention guidance to ensure temporal consistency and avoid semantic entanglement.

A Frame-wise Information Gain Principle (FIGP) to ensure high-quality token updates.

Dense frame sampling and explicit inclusion of occluded frames during training to enhance robustness.

The method achieves state-of-the-art performance on multiple benchmarks (LaSOT, LaSOText, GOT-10K, TrackingNet, UAV123, TNL2K) while maintaining high inference speed.

**Strengths:**

1. State-of-the-Art Performance Across Diverse and Challenging Benchmarks
STARTRACK does not merely achieve top results on one or two benchmarks; it demonstrates generalized superiority across a wide spectrum of challenges, which is a strong indicator of its robustness.

LaSOT & LaSOText: Achieving 75.2% AUC on LaSOT and 53.2% on LaSOText is significant because these are large-scale, long-term benchmarks. LaSOText, in particular, with its focus on similar distractors and frequent occlusions, directly validates the core claim that STARTRACK excels at spatio-temporal modeling and discrimination beyond simple appearance matching.

GOT-10K: The high score of 78.5% AO under the strict one-shot protocol (training only on GOT-10K's training split) is a powerful testament to the model's generalization capability. This shows that the learned representation evolution strategy is not overfitted to specific object categories.

UAV123 & TNL2K: Superior performance on UAV123 (aerial perspective, small objects, fast motion) and TNL2K (diverse media including cartoons) proves the framework's adaptability to different domains and data sources, moving beyond conventional RGB video.

2. A Novel and Paradigm-Shifting Pipeline: From Matching to Evolution
The most profound contribution is the conceptual shift from a static matching paradigm to a dynamic evolution paradigm.

Beyond Template Degradation: Traditional trackers, even those with dynamic template updates, fundamentally perform matching. STARTRACK abandons this entirely. Its iterative token propagation mechanism allows the tracker to build a continuously updating "memory" of the target and its relationship with the environment. This explicitly mitigates the Achilles' heel of long-term tracking: the gradual irrelevance of the initial template.

Token as a Dynamic State Vector: The target and context tokens act as a compact, learned state vector that carries all necessary historical information. This is more elegant and potent than hand-crafted update strategies for multiple templates or complex cross-frame attention mechanisms, leading to a simpler yet more effective architecture.

3. A Holistic and Innovative Training Strategy

Dense Sampling with Occlusion: By using densely sampled sequences and, for the first time, explicitly including occluded frames in training, the model is forced to learn a crucial skill: reasoning without appearance. This trains the context tokens to capture the underlying spatial structure and motion patterns of the scene, enabling the tracker to hypothesize the target's location even when it is invisible.

Frame-wise Information Gain Principle (FIGP): This is a clever solution to a key problem in propagation-based models: ensuring that each update is beneficial. FIGP provides a self-supervised, internal consistency signal that actively prevents representation collapse and encourages the tokens to become progressively more informative, ensuring stable long-term performance.

**Weaknesses:**

1. Insufficient Depth in Related Work on Temporal Modeling
The paper's review of existing temporal methods is somewhat narrow, missing a discussion of several influential works that would provide a richer context for its contributions. PrDiMP (Probabilistic Regression and DIMP)，STMTrack (Spatio-Temporal Memory Trackers)，TCTrack (Temporal Context Trackers)，MeMOTR (Memory-Augmented MOT with Transformers)

2. Limited Analysis of Failure Modes and Robustness Boundaries
The paper convincingly demonstrates success but offers less insight into its limitations.

Extreme Deformation or Fast Motion: How does the token propagation mechanism cope when the target undergoes radical non-rigid deformation or moves so fast that its appearance changes drastically between frames? The dense sampling may help, but the upper limits are not explored.

Full Scene Changes: What happens when the camera cuts to a completely different scene (a common challenge in long-term TV show tracking)? The reliance on spatio-temporal context would likely break down, and it's unclear how the model would recover.

Initialization Sensitivity: While an ablation on target token initialization is provided, a deeper analysis of how sensitive the entire tracking process is to errors or noise in the initial bounding box is missing.

3. Practical Deployment Considerations

Computational Cost of Dense Sampling: While the inference FPS is high, the training cost of using densely sampled sequences is significantly higher than sparse sampling. The paper does not discuss the computational overhead of this training strategy.

Hyperparameter Sensitivity: The performance appears sensitive to hyperparameters like the token matrix size (Fig. 5) and sampling length (Fig. 6). This suggests that optimal deployment on a new dataset might require non-trivial tuning, potentially limiting its applicability.

**Questions:**

1. Insufficient Depth in Related Work on Temporal Modeling
2. Limited Analysis of Failure Modes and Robustness Boundaries
3. Practical Deployment Considerations

---

> ### Author Response · Authors · 2025-11-26
> **Reply to Reviewer cD5U (1/3)**
>
> Thank you for your valuable feedback, which has helped us improve the paper. We have made revisions based on your comments. Below are the details of our changes. Please let us know if we have addressed your concerns.
>
> >**Q1:** Insufficient Depth in Related Work on Temporal Modeling The paper's review of existing temporal methods is somewhat narrow, missing a discussion of several influential works that would provide a richer context for its contributions. PrDiMP (Probabilistic Regression and DIMP), STMTrack (Spatio-Temporal Memory Trackers),TCTrack (Temporal Context Trackers), MeMOTR (Memory-Augmented MOT with Transformers)
>
> **A1:** Thank you for your valuable feedback. We acknowledge that our discussion of existing temporal modeling methods could have been more comprehensive. The works you mentioned, such as STMTrack, TCTrack, and MeMOTR, are indeed influential and contribute significantly to the field of temporal modeling for tracking. MeMOTR jointly modeling multiple search frames for cross-frame context association, while STMTrack and TCTrack model multi-frame associations by introducing additional spatio-temporal modules. We have revised the paper and added these details to Section 2.1.
>
> Unfortunately, we checked the PrDiMP model (Danelljan M, Gool L V, Timofte R. Probabilistic regression for visual tracking[C]//Proceedings of the IEEE/CVF conference on computer vision and pattern recognition. 2020: 7183-7192), and found that it still follows the image-matching paradigm and does not introduce temporal modeling. We would greatly appreciate it if you could provide the specific paper or reference you had in mind.
>
>
> >**Q2:** Extreme Deformation or Fast Motion: How does the token propagation mechanism cope when the target undergoes radical non-rigid deformation or moves so fast that its appearance changes drastically between frames? The dense sampling may help, but the upper limits are not explored.
>
> **A2:**
>  - **Extreme Deformation.** Extreme deformation refers to significant changes in the appearance of the object. In fact, our training set includes many cases of object deformation, and the target representation tokens can effectively learn the changes in the object's appearance during the training phase. Fig. 5 shows the AUC comparison of 15 challenging scenarios on LaSOT, where STARTrack achieved the highest performance at 76.4\% in handling object deformation.
>  - **Fast Motion.** Fast motion typically refers to sudden jumps in the target's position between consecutive frames, often accompanied by motion blur. In such cases, relying on historical motion trajectories is unreliable, and we attribute the solution to appearance matching. While image matching paradigms seem to handle this well, STARTrack still achieved a high AUC of 63.6\% in fast motion scenarios (see Fig. 5). We attribute its excellent performance to the introduction of context representation tokens. These tokens, leveraging historical context and guided by negative attention, effectively filter out background regions and highlight potential target locations, thus guiding accurate target localization. This is where image matching paradigms have limitations. Additionally, in Section 6, we provide attention visualizations of both the target representation tokens and context representation tokens in motion blur scenarios, further supporting our point.

---

> ### Author Response · Authors · 2025-11-26
> **Reply to Reviewer cD5U (2/3)**
>
> **Q3:** Full Scene Changes: What happens when the camera cuts to a completely different scene (a common challenge in long-term TV show tracking)? The reliance on spatio-temporal context would likely break down, and it's unclear how the model would recover.
>
> **A3:** We appreciate your attention. Unfortunately, the existing test sequences are based on continuous scenes, and we have not found any relevant analysis on the challenge of scene change. We agree with your point that scene changes can lead to the failure of spatiotemporal dependencies, and target position prediction through trajectory becomes impossible. The re-capture of the target can only rely on its appearance. We believe that the image matching paradigm is sufficient to handle this challenge, and the target representation tokens in STARTrack can accurately re-capture the target.
>
>
> >**Q4:** Initialization Sensitivity: While an ablation on target token initialization is provided, a deeper analysis of how sensitive the entire tracking process is to errors or noise in the initial bounding box is missing.
>
> **A4:** We add an ablation experiment on the impact of different initializations of target representation tokens on performance, as shown in Tab. 1. Without any numerical enhancement of the target region, the target representation tokens can hardly perceive the initial state of the target. We further compared three different enhancement strategies: constant enhancement, Gaussian edge attenuation, and bounding-box-guided target segmentation. All three approaches enabled the model to perceive the target region, resulting in comparable performance across the three (&#177;0.4%). For the sake of framework simplicity and efficient reference frame preprocessing, we adopted constant enhancement as the default strategy. We have revised the paper and added these details to Section 5.3.5.
>
> **Tab. 1** Comparison of different target enhancement methods on the reference frame for target representation token initialization.
>
> | Target Enhancement    | LaSOT $ \uparrow $ | UAV123 $ \uparrow $ | TNL2K $ \uparrow $ |
> |-----------------------|-------|--------|-------|
> | N/A                   | 25.2  | 27.3   | 19.6  |
> | Constant              | 73.8 | **71.9** | **60.3** |
> | 2D Gaussian           | 73.5  | 72.2   | 59.9  |
> | Segment & Constant    | **74.1**  | 71.8   | 60.0  |

---

> ### Author Response · Authors · 2025-11-26
> **Reply to Reviewer cD5U (3/3)**
>
> **Q5:** Computational Cost of Dense Sampling: While the inference FPS is high, the training cost of using densely sampled sequences is significantly higher than sparse sampling. The paper does not discuss the computational overhead of this training strategy.
>
> **A5:** We understand your point and have re-examined the relevant code. After further analysis and experimentation, we believe that the training cost of dense sequence sampling is equivalent to that of sparse sampling. We have adopted the sampler from traditional tracking frameworks, which reads all the frame numbers of a random sequence and performs sampling. The selected frame numbers are mapped to the image frame paths of the video sequence, allowing the dataloader to read the image frames according to the paths and perform subsequent image augmentation operations. Given the specified number of training frames, the difference between the two sampling methods lies in how the specific frames are selected.
>
>
> >**Q6:** Hyperparameter Sensitivity: The performance appears sensitive to hyperparameters like the token matrix size (Fig. 5) and sampling length (Fig. 6). This suggests that optimal deployment on a new dataset might require non-trivial tuning, potentially limiting its applicability.
>
> **A6:** In our tracking process, the search frames are cropped and resized based on the target bounding box, leading to relatively fixed target and context representations. Therefore, we believe that on a new dataset, as long as the search area processing method remains consistent, there is no need for additional adjustment of the token matrix size.
>
> Due to GPU memory limitations, we kept the batch size at 40 for the ablation study on different sampling lengths, using this setup to evaluate the impact of sequence count and sampling length on model performance. As shown in Figure 6, when the sampling length $\leq$ 10, the model performance gradually improves and remains at a high level (73.8\% AUC on LaSOT, 77.5\% AUC on GOT-10k), despite a slight decrease in the number of sequences. We believe that higher experimental configurations and longer sequence sampling can enable the model to reach higher performance levels.

---

### Official Review · Reviewer_LieF · 2025-10-31

**Soundness:** 2
**Presentation:** 2
**Contribution:** 2
**Rating:** 4
**Confidence:** 5

**Summary:**

This work presents a simple yet effective video-level tracking framework termed STARTrack, which realizes the temporal evolution of target and context representations using an iterative token propagation mechanism. It takes two types of tokens that carry historical representations to update the static target templates over time for target state inference. The tracker has been evaluated on various public benchmarks.

**Strengths:**

1. The writing is clear and method is easy to follow. The introduction of two distinct types of tokens to decouple and model the target's appearance and environmental relationship seems reasonable.
2. The paper provides extensive quantitative results on the public tracking benchmarks, demonstrating the method's robustness, and efficiency.

**Weaknesses:**

1. The strategy of using dynamic template tokens to propagate temporal information is not very novel. Numerous prior works like HIPTrack, SPMTrack also employ token storage and temporal information propagation techniques to improve the performance. The proposed dual-stream token mechanism is an incremental combination of existing ideas rather than a groundbreaking new paradigm.
2. The performance gain compared to SPMTrack in Table 1 is marginal.
3. The in-depth analyses of the two types of tokens have not been clearly provided.

**Questions:**

1. Could the authors provide more detail analyses or visualizations of the two types of tokens?
2. What’s the reason that the AUC drops significantly (from 73.8% to 44.7%) when the Frame-wise Information Gain Principle (FIGP) is removed?

---

> ### Author Response · Authors · 2025-11-26
> **Reply to Reviewer LieF (1/2)**
>
> Thank you for your valuable feedback, which has helped us improve the paper. We have made revisions based on your comments. Below are the details of our changes. Please let us know if we have addressed your concerns.
>
> >**Q1:** The strategy of using dynamic template tokens to propagate temporal information is not very novel. Numerous prior works like HIPTrack, SPMTrack also employ token storage and temporal information propagation techniques to improve the performance. The proposed dual-stream token mechanism is an incremental combination of existing ideas rather than a groundbreaking new paradigm.
>
> **A1:** We agree with your summary of prior works, but these methods still rely on the continuous involvement of the initial template $R_0$ throughout the tracking process. In contrast, our proposed STARTrack completely eliminates the dependency on the initial template and adaptively models the target's spatiotemporal cues and the contextual spatiotemporal dynamics. This greatly enhances the tracker's long-term tracking performance, as long-term tracking often involves significant changes in the target's appearance.
>
>
> >**Q2:** The performance gain compared to SPMTrack in Tab. 1 is marginal.
>
> **A2:** SPMTrack uses a more powerful ViT pre-training, namely DINOv2, while STARTrack use a ViT encoder pre-trained with MAE. In our paper, we have updated the performance of the DINOv2 version of STARTrack, labeled as STARTrack-D. Tab. 1 and Tab. 2 shows the performance comparison between STARTrack-D and other state-of-the-art trackers. STARTrack-D256 and STARTrack-D384 achieve excellent performance compared to trackers with the same resolution, with AUC scores of 74.4\% and 76.0\%, respectively. We have revised the paper and added these details to Section 5.2.
>
> **Tab. 1** Comparison with state-of-the-art trackers on four popular benchmarks: LaSOT, $\rm LaSOT_{ext}$, GOT-10K, and TrackingNet. Where * denotes for trackers only trained on GOT-10K. Best in bold. Tracker resolution: 256×256.
> | Method              | LaSOT AUC |  **$\rm LaSOT_{ext}$** AUC | GOT-10K* AO | TrackingNet AUC |
> |---------------------|:-----------------------:|:-----------------:|:----------------:|:-----------------:|
> | **STARTrack-D256**   | **74.4**  | **52.5**             | **78.0**       | **74.4**        |
> | STARTrack-256    | 73.8      | 52.1                 | 77.8           | 74.1            |
> | ARPTrack-256         | 72.6      | 52.0                 | 77.7           | 74.3            |
> | ARTrackV2-256        | 71.6      | 50.8                 | 75.9           | 72.7            |
> | AQATrack-256         | 71.4      | 51.2                 | 73.8           | 72.1            |
> | ROMTrack-256         | 69.3      | 48.9                 | 72.9           | 70.2            |
> | ARTrack-256          | 70.4      | 46.4                 | 73.5           | 70.9            |
> | OSTrack-256          | 69.1      | 47.4                 | 71.0           | 68.2            |
> | MixFormer-22k        | 69.2      | -                    | 70.7           | 67.8            |
>
> **Tab. 2** Comparison with state-of-the-art trackers on four popular benchmarks: LaSOT, $\rm LaSOT_{ext}$, GOT-10K, and TrackingNet. Where * denotes for trackers only trained on GOT-10K. Best in bold. Tracker resolution: 384×384.
> | Method              | LaSOT AUC |  **$\rm LaSOT_{ext}$** AUC | GOT-10K* AO | TrackingNet AUC |
> |---------------------|:-----------:|:-----------------:|:----------------:|:-----------------:|
> | **STARTrack-D384**   | **76.0**  | **53.6**             | **79.0**       | **77.1**        |
> | STARTrack-384        | 75.2      | 53.2            | 78.5       | 76.5        |
> | SPMTrack-384         | 74.9      | -                    | 76.5           | 76.3            |
> | ODTrack             | 73.2      | 52.4                 | 77.0           | 75.1            |
> | ARTrackV2-384       | 73.0      | 52.9                 | 77.5           | 75.5            |
> | ARTrack-384         | 72.6      | 51.9                 | 75.5           | 74.3            |
> | ROMTrack-384        | 71.4      | 51.3                 | 74.2           | 72.4            |
> | SeqTrack-B384       | 71.5      | 50.5                 | 74.5           | 71.4            |
> | OSTrack-384         | 71.1      | 50.5                 | 73.7           | 70.8            |

---

> ### Author Response · Authors · 2025-11-26
> **Reply to Reviewer LieF (2/2)**
>
> >**Q3:** The in-depth analyses and visualizations of the two types of tokens have not been clearly provided.
>
> **A3:** Thank you for your suggestion. We believe that visualizing the target representation tokens and context representation tokens during the tracking process can more directly demonstrate their roles, thereby making the entire framework more complete. In Section 6, we present challenging scenarios involving motion blur and occlusion and visualize the attention distributions of both types of tokens. Notably, these scenes contain many similar-looking objects. Although in such cases, our target representation tokens occasionally attend to incorrect instances, the context representation tokens can accurately determine the target's location by leveraging dense spatio-temporal correlations. We have revised the paper and added these details to Section 6.
>
>
> >**Q4:** What's the reason that the AUC drops significantly (from 73.8\% to 44.7\%) when the Frame-wise Information Gain Principle (FIGP) is removed?
>
> **A4:** The Frame-wise Information Gain Principle (FIGP) is the core of STARTrack's training strategy. It requires representation tokens to efficiently fuse search frame features and reduce noise introduction. Specifically, for a given search frame $S_{T+1}$, representation tokens $\\{\phi^{tgt}\_{T}, \phi^{ctx}\_{T}\\}$ are expected to capture richer spatio-temporal information than $\\{\phi^{tgt}\_{T-1}, \phi^{ctx}\_{T-1}\\}$, thus enabling more accurate predictions. Due to computational resource limitations, we train the model using only a 5-frame short sequence. Without using FIGP, the tracker can only maintain stable tracking for the initial few frames during testing, and its robustness declines in the subsequent tracking process due to noise interference. However, the tracker trained with FIGP allows representation tokens to be efficiently updated and effectively constrains noise interference, leading to more robust tracking.

---

### Official Review · Reviewer_i6rU · 2025-11-01

**Soundness:** 3
**Presentation:** 3
**Contribution:** 3
**Rating:** 6
**Confidence:** 4

**Summary:**

STARTrack is a video-level visual object tracking framework that departs from static template-matching paradigms by explicitly modeling the spatio-temporal evolution of both target and context representations. The method uses an iterative token-propagation design: two kinds of learnable representation tokens (target tokens and context tokens) are propagated across frames and fused with current-frame visual features through a dual-stream causal attention mechanism. The model also incorporates negative attention guidance to encourage context tokens to focus on spatio-temporally relevant cues. To support stable temporal updates, STARTrack introduces a tailored training strategy centered on the Frame-wise Information Gain Principle (FIGP), dense sampling, and occlusion-aware training; losses include a temporal refinement term that enforces that current-frame predictions should improve (or not be worse than) those from previous tokens. Experiments on multiple benchmarks (LaSOT, GOT-10K, TrackingNet, TNL2K, UAV123, LaSOText) report state-of-the-art accuracy while maintaining competitive inference speed. Ablations validate the benefits of dual token types, dense sampling, FIGP, and occlusion inclusion during training.

**Strengths:**

- Iterative token-propagation for spatio-temporal representation evolution: STARTrack proposes propagating learned target and context representation tokens across frames and fusing them with current-frame features. This token-centric propagation explicitly models representation evolution over time and reduces dependence on a fixed initial template, addressing long-term drift.

- Dual-stream causal attention with negative attention guidance: The architecture separates modeling of target and context into parallel causal attention streams to avoid mutual interference, and introduces negative attention guidance to help context tokens converge faster and focus on discriminative spatio-temporal context relevant to the tracked object.

- Training innovations for temporal consistency and occlusion robustness: They formulate the Frame-wise Information Gain Principle and add a temporal refinement term to losses that enforces improvement (or non-degradation) of current predictions relative to previous tokens. They also densely sample frames and explicitly include occluded frames in training, helping the model learn context-driven localization when appearance cues are weak.

**Weaknesses:**

- Complexity and interpretability of token dynamics: While token propagation is powerful, the paper relies on many design choices (two token types, negative guidance sign flips, dual-stream masking/ordering, initialization strategies). These choices introduce complexity; ablation shows sensitivity (e.g., FIGP removal causes large drops), but the conceptual interpretability and theoretical understanding of token dynamics and stability over very long sequences remain limited.
- Dependence on many engineered training choices and hyperparameters: The method’s strong performance hinges on several specific training strategies (dense sampling, occlusion inclusion, FIGP temporal loss terms, representation initialization variants). This could make reproducibility or transfer to other datasets/domains sensitive to hyperparameters and dataset composition; the paper indicates some instability when modifying multi-frame variants.
- Limited analysis of failure cases and computational trade-offs in diverse settings: Although STARTrack reports good FPS and benchmark numbers, the paper contains limited discussion of failure modes (e.g., heavy crowding, severe distractors, extreme viewpoint/scale changes) and how the token mechanism behaves in such scenarios. Also, while the framework avoids explicit motion modules to keep inference fast, more explicit comparison of compute/memory cost vs. competing long-term trackers (especially under very long sequences or constrained hardware) would strengthen practical claims.

**Questions:**

SEE WEAKNESS

---

> ### Author Response · Authors · 2025-11-26
> **Reply to Reviewer i6rU (1/3)**
>
> Thank you for your valuable feedback, which has helped us improve the paper. We have made revisions based on your comments. Below are the details of our changes. Please let us know if we have addressed your concerns.
>
> >**Q1:** Complexity and interpretability of token dynamics: While token propagation is powerful, the paper relies on many design choices (two token types, negative guidance sign flips, dual-stream masking/ordering, initialization strategies). These choices introduce complexity; ablation shows sensitivity (e.g., FIGP removal causes large drops), but the conceptual interpretability and theoretical understanding of token dynamics and stability over very long sequences remain limited.
>
> **A1:** We appreciate your attention and would like to further explain the motivations behind these heuristic design choices and their effects. Although these design choices (such as two token types, negative guidance sign flip, dual-stream masking/ordering, and initialization strategies) do add complexity to the model, they also play a key role in enhancing the flexibility and performance of the model.
>  - **Two token types:** The target representation token and context representation token focus on the target's spatiotemporal cues and scene changes from different perspectives, helping the model better locate the target in complex environments, especially in cases of occlusion or similar objects.
>  - **Negative guidance sign flip:** This design introduces a "reverse attention" mechanism to help the model maintain consistency in the target bounding box during training. While this adds complexity to the design, it improves the model's robustness to uncertainty (such as occlusion and target changes). Please refer to Section 5.3.2 for detailed results.
>  - **Dual-stream masking/ordering:** Through the dual-stream structure, the model can separately handle target and background information, preventing information confusion. This helps enhance the model's discriminative ability, especially for tracking tasks in complex scenes.
>  - **Initialization strategy:** Since we cannot directly obtain initial target information from the reference frame, we provide the model with initial target position hints by weighting the target bounding box within the reference frame. This is a simple yet effective strategy that avoids excessive reliance on complex external annotations.
>
>
> We agree that the interpretability and stability of token dynamics in very long sequences remain areas that need further research, as our token update mechanism is fully adaptive. But the Frame-wise Information Gain Principle (FIGP) is the core of STARTrack's training strategy. It requires representation tokens to efficiently fuse search frame features and reduce noise introduction. Specifically, for a given search frame $S_{T+1}$, representation tokens $\\{\phi^{tgt}\_{T}, \phi^{ctx}\_{T}\\}$ are expected to capture richer spatio-temporal information than $\\{\phi^{tgt}\_{T-1}, \phi^{ctx}\_{T-1}\\}$, thus enabling more accurate predictions. Due to computational resource limitations, we train the model using only a 5-frame short sequence. Without using FIGP, the tracker can only maintain stable tracking for the initial few frames during testing, and its robustness declines in the subsequent tracking process due to noise interference. However, the tracker trained with FIGP allows representation tokens to be efficiently updated and effectively constrains noise interference, leading to more robust tracking. Therefore, we believe that FIGP is an essential training method highly adapted to STARTrack, rather than an inherent sensitivity of STARTrack itself.

---

> ### Author Response · Authors · 2025-11-26
> **Reply to Reviewer i6rU (2/3)**
>
> >**Q2:** Dependence on many engineered training choices and hyperparameters: The method's strong performance hinges on several specific training strategies (dense sampling, occlusion inclusion, FIGP temporal loss terms, representation initialization variants). This could make reproducibility or transfer to other datasets/domains sensitive to hyperparameters and dataset composition; the paper indicates some instability when modifying multi-frame variants.
>
> **A2:** Thank you for your attention to our training strategy. STARTrack introduces a new paradigm in the tracking field, which is fully adaptive in capturing the spatio-temporal dynamics of the target, rather than being limited to the initial appearance of the target, $R_0$. Sequence-level dense sampling has been widely used in the training of autoregressive trackers, and the introduction of occlusion frames effectively enhances the discriminative power of context tokens for target-related spatio-temporal context, while allowing target representation tokens to maintain the target representation when the target is lost, thereby reducing noise introduction. To impose constraints on the tracker training with the Frame-wise Information Gain Principle (FIGP) strategy, we designed an additional loss term, which is in fact a part of the FIGP strategy. Specifically, target representation tokens, after being fused with more search frame features, should contain richer temporal information for more accurate predictions.
>
> We also conduct an ablation experiment on the impact of different initializations of target representation tokens on performance, as shown in Tab. 1. Without any numerical enhancement of the target region, the target representation tokens can hardly perceive the initial state of the target. We further compared three different enhancement strategies: constant enhancement, Gaussian edge attenuation, and bounding-box-guided target segmentation. All three approaches enabled the model to perceive the target region, resulting in comparable performance across the three (&#177;0.4%)
> . For the sake of framework simplicity and efficient reference frame preprocessing, we adopted constant enhancement as the default strategy. We have revised the paper and added these details to Section 5.3.5. We believe these training strategies are simple, intuitive, easy to implement, and align with our framework design philosophy. Additionally, they exhibit generalizability, making them easy to transfer to other datasets/domains.
>
> **Tab. 1** Comparison of different target enhancement methods on the reference frame for target representation token initialization.
>
> | Target Enhancement    | LaSOT $ \uparrow $ | UAV123 $ \uparrow $ | TNL2K $ \uparrow $ |
> |-----------------------|-------|--------|-------|
> | N/A                   | 25.2  | 27.3   | 19.6  |
> | **Constant**            | 73.8 | **71.9** | **60.3** |
> | 2D Gaussian           | 73.5  | 72.2   | 59.9  |
> | Segment & Constant    | **74.1**  | 71.8   | 60.0  |
>
>
> The instability observed when changing the multi-frame variant (AUC-0.3\%) is not due to the training strategy but rather a limitation of the Multi-frame-wise Information Gain Principle. This principle only focuses on the information gain between the last frame and the first frame, without imposing strict constraints on the token evolution of intermediate frames, which leads to a drop in accuracy.

---

> ### Author Response · Authors · 2025-11-26
> **Reply to Reviewer i6rU (3/3)**
>
> >**Q3:** Limited analysis of failure cases and computational trade-offs in diverse settings: Although STARTrack reports good FPS and benchmark numbers, the paper contains limited discussion of failure modes (e.g., heavy crowding, severe distractors, extreme viewpoint/scale changes) and how the token mechanism behaves in such scenarios. Also, while the framework avoids explicit motion modules to keep inference fast, more explicit comparison of compute/memory cost vs. competing long-term trackers (especially under very long sequences or constrained hardware) would strengthen practical claims.
>
> **A3:** Although STARTrack achieves excellent performance on various benchmark datasets, its tracking capability is still limited in extremely complex scenes. For example, in cases of long-term occlusion or when the target exits and re-enters the field of view after some time, the tracker fails to capture the target's appearance and motion trajectory. As a result, continuous modeling of the background region introduces slight noise into the target representation token, which reduces its representation quality, leading to tracking drift or mis-tracking of similar objects. A feasible solution is to introduce an additional mechanism for detecting target disappearance. Furthermore, due to GPU memory limitations, STARTrack trains with a batch size of 8 sequences, each containing 5 consecutive frames, which somewhat restricts the ability of representation tokens to propagate over long periods. With richer sampling sequences, better hardware configurations, and longer sampling sequences, STARTrack's long-term tracking potential can be more effectively unlocked. We have revised the paper and added the above limited analysis to Appendix F.
>
> Tab. 2 compares the computational and memory costs of STARTrack with other long-term trackers. Since STARTrack requires neither additional reference frame inputs nor complex module designs, it achieves an inference speed of 84 FPS at a resolution of 384, which is nearly 2 times faster than AQATrack-384. It is worth noting that STARTrack-D, which uses the same backbone as SPMTrack, has fewer parameters and lower computational cost, which results in higher inference speed. We have revised the paper and added these details to Section 5.1.
>
> **Tab. 2** Comparison of model parameters, FLOPs, and inference speed, with tracker resolution unified to 384×384.
>
> | Method        | Type        | Params $ \downarrow $ | FLOPs $ \downarrow $ | Speed $ \uparrow $  | Device  |
> |---------------|-------------|--------|-------|--------|---------|
> | SeqTrack      | ViT-B       | 89 M   | 148 G | 11 fps | 2080Ti  |
> | ODTrack       | ViT-B       | 92 M   | 73 G  | 32 fps | 2080Ti  |
> | AQATrack      | HiViT-B     | 72 M   | 58 G  | 44 fps | v100    |
> | SPMTrack      | DINOv2-B    | 115 M  | 86 G  | 26 fps | 2080Ti  |
> | **STARTrack** | **ViT-B**   | **86 M** | **36 G** | **84 fps** | 2080Ti |
> | STARTrack-D| DINOv2-B | 92 M | 48 G | 70 fps | 2080Ti |

---

### Official Review · Reviewer_iYMN · 2025-11-03

**Soundness:** 1
**Presentation:** 1
**Contribution:** 2
**Rating:** 2
**Confidence:** 5

**Summary:**

This paper proposes a new method to tackle object tracking problem. The proposes method introduces a few new designs in the network architecture to enhance the performance of object tracking. It adopts ViT as visual encoder to embed the input images, together with historic tokens into embeddings. Secondly, it introduces extra query tokens that helps to decouple semantic features. It also adopts dual stream attention mechanism to learn discriminative features from targets and contexts. Extensive experiments show that the proposed method outperforms baseline methods on the benchmark quantitatively.

**Strengths:**

1. The quantitative results show that the proposed method performs the best compared to baseline methods.

**Weaknesses:**

1. What are $H_1, H_2, ... H_n$ in Eq.(2) ?   $H_i$ and $W_i$  are used to represent both image size and network parameters in a confusing way.
2. What do symbols $q, k, v$ represent in Eq. (2)?  How do they differ from $Q, K, V$?
3. Where are the dual stream attention shown in the Figure 2?  It's difficult to understand how the dual stream work in the entire network architecture.
4. Section 3.3.2 Why do the query tokens avoid semantic entanglement? How do they achieve this goal?
5. The experiments do not have any qualitative results.

**Questions:**

The main issue of this paper submission is the presentation and the development of the paper content.  The organization and presentation of this manuscript is poor such that it's hard to follow the logic flow. The mathematical symbols are not properly defined and explained progressively. The figures and diagrams are not properly referred from the text.  The authors are suggested to paraphrase and polish the manuscript text carefully.

---

> ### Author Response · Authors · 2025-11-26
> **Reply to Reviewer iYMN (1/2)**
>
> Thank you for your valuable feedback, which has helped us improve the paper. We have made revisions based on your comments. Below are the details of our changes.
>
> >**Q1:** What are $H_1, H_2, \dots, H_n$ in Eq.(2)? $H_i$ and $W_i$ are used to represent both image size and network parameters in a confusing way.
>
> **A1:** We apologize for any confusion caused by our previous explanation. In fact, in Eq. (2), we are demonstrating multi-head attention with 12 heads. $H_i$ represents the output of the $i$-th attention head, which is computed as $\text{Attention}(QW_{i}^{Q}, KW_{i}^{K}, V_G W_{i}^{V})$. $W_i$ represents the learnable weight matrix, which is used during the computation to apply a linear transformation to the input query ($Q$), key ($K$), and value ($V$) vectors. To avoid confusion with the notation for the input image size $S_T \in \mathbb{R}^{H_s \times W_s \times 3}$, we have revised Eq.(2) and its explanation as follows:
>
> The multi-head attention in layer $j$ is computed as follows:
>
> $$
> \begin{equation*}
> \begin{aligned}
> &\text{MultiHead}(\mathbf{Q}, \mathbf{K}, \mathbf{V}) = \text{Concat}(\mathbf{H}_1, \cdots, \mathbf{H}_n) \mathbf{W}^O, \\\\
> &\mathbf{H}_i = \text{Attention}(\mathbf{Q} \mathbf{W}_i^Q, \mathbf{K} \mathbf{W}_i^K, \mathbf{V}_G \mathbf{W}_i^V), \\\\
> &\text{Attention}(\mathbf{q}, \mathbf{k}, \mathbf{v}) = \text{softmax}(\frac{\mathbf{q} \mathbf{k}^T}{\sqrt{d_k}} + \text{AttnMask}) \mathbf{v}.
> \end{aligned}
> \end{equation*}
> $$
>
>
> Here, $\mathbf{W}_i^Q \in \mathbb{R}^{d_m \times d_k}$, $\mathbf{W}_i^K \in \mathbb{R}^{d_m \times d_k}$, $\mathbf{W}_i^V \in \mathbb{R}^{d_m \times d_v}$, and $\mathbf{W}^O \in \mathbb{R}^{n \times d_m}$ are learnable parameters. In our case, we use $\mathbf{Q} = \mathbf{K} = \text{Concat}(f_s^j; \phi^{tgt,j}\_{T-1}, \phi^{ctx,j}\_{T-1}, \phi^{tgt,j}\_q, \phi^{ctx,j}\_q)$, $\mathbf{V}_G = \text{Concat}(f_s^j; \phi^{tgt,j}\_{T-1}, -\phi^{ctx,j}\_{T-1}, \phi^{tgt,j}\_q, -\phi^{ctx,j}\_q)$, $n = 12$, $d_m = 768$, and $d_k = d_v = d_m / n = 64$.
>
> We have revised the paper and added these details to Section 3.2.
>
> >**Q2:** What do symbols $q, k, v$ represent in Eq. (2)? How do they differ from $Q, K, V$?
>
> **A2:** In Eq.(2), the third equation provides a detailed explanation of the Attention operator. To avoid ambiguity, we use $ \mathbf{q}, \mathbf{k}$ and $\mathbf{v}$ to represent the query, key, and value vectors, respectively, which helps distinguish them from the matrices in the second equation.
>
> In the second equation of Eq. (2),
> $\mathbf{q} = \mathbf{Q} \mathbf{W}_i^Q, \mathbf{k} = \mathbf{K} \mathbf{W}_i^K, \mathbf{v} = \mathbf{V}_G \mathbf{W}_i^V.$ We hope this explanation can clarify the distinction and address your concern.

---

> ### Author Response · Authors · 2025-11-26
> **Reply to Reviewer iYMN (2/2)**
>
> >**Q3:** Where are the dual stream attention shown in Figure 2? It's difficult to understand how the dual stream work in the entire network architecture.
>
> **A3:**  We apologize that the wording in our manuscript did not help you understand how the dual stream works in the entire network architecture. The dual-stream attention is illustrated in Fig.2(d).
>
> As described in Eq. (2), we control the direction of information flow by adding an attention score mask to the self-attention matrix, in order to restrict the direct interaction between $ \\{\phi^{tgt}\_{T-1}, \phi^{tgt}\_{q}\\} $ and $ \\{\phi^{ctx}\_{T-1}, \phi^{ctx}\_{q}\\} $. Fig. 2(d) illustrates the specific attention directions: the attention interaction between $ \\{\phi^{tgt}\_{T-1}, \phi^{tgt}\_{q}\\} $ and $ f_{s}^{0} $ in the visual encoder performs target spatio-temporal feature updating and fusion, while the attention interaction between $ \\{\phi^{ctx}\_{T-1}, \phi^{ctx}\_{q}\\} $ and $ f_{s}^{0} $ is responsible for capturing the dynamic patterns of the context features. By decoupling the inference into parallel streams, the model maintains the independence of different semantic sources and prevents mutual interference during feature aggregation, thereby generating more robust and interpretable spatio-temporal representations.
>
> We hope this explanation clarifies your confusion, and we have revised the paper and added these details to Section 3.3.2.
>
> >**Q4:** Section 3.3.2: Why do the query tokens avoid semantic entanglement? How do they achieve this goal?
>
> **A4:** In Section 3.3, we introduced token queries. The token queries $\\{\phi^{tgt,j}\_{q}, \phi^{ctx,j}\_{q}\\}$ are responsible for adaptively aggregating current search frame features and historical representations from the representation tokens, and storing the fused features. The specific process is shown in Fig. 2(d) in our paper. Taking $\phi^{tgt,j}\_{q}$ as an example, it uses the attention mechanism to focus on $\phi^{tgt,j}\_{T-1}$ and search frame features $f_{s}^{j+1}$ separately, performing token-level adaptive fusion and storage. Without the introduction of token queries, the temporal feature update can only rely on the target representation token $\phi^{tgt,j}\_{T-1}$, where the target representation token $\phi^{tgt,j}\_{T-1}$ and the search frame features $f_{s}^{j+1}$ are computed using bidirectional standard attention, instead of causal attention, which leads to temporal semantic entanglement. You can refer to the design in Tab. 1 below or Fig. 3(b) in our paper, as this approach results in a 2.7% performance drop.
>
> **Tab. 1** Comparison on different combination of representation tokens on LaSOT.
>
> | Token Combination    | AUC $ \uparrow $   | $\rm P_{norm}$ $ \uparrow $ | P $ \uparrow $    |
> |----------------------|-------|----------------|------|
> | **STARTrack**            | **73.8** | **83.5**       | **80.4** |
> | w/o target tokens    | 72.4  | 81.9           | 78.5 |
> | w/o context tokens   | 69.5  | 79.2           | 76.0 |
> | w/o token queries    | 72.1  | 82.0           | 77.9 |
>
>
> >**Q5:** The experiments do not have any qualitative results.
>
> **A5:** Thank you for your attention. We have added attention visualizations of the target representation tokens and the context representation tokens in the manuscript to intuitively demonstrate the effectiveness of our proposed method. We have revised the paper and added these details to Section 6.
>
> Thank you for your attention to our work. We have thoroughly revised and refined the entire paper to enhance its logical coherence and readability. Please let us know if we have adequately addressed your concerns.

---

### Author Response · Authors · 2025-12-01
**Global response (2/2)**

## Summary of Rebuttal

### Reviewer iYMN

We notice that there may be some factual misunderstandings regarding our work, and we hope that the following revisions will help you better understand our work.


1. **Clarification of Equations**: We revised Equation (2) to clarify the distinction between $H_i$ (attention head outputs) and $W_i$ (weight matrices). This addresses confusion in our previous explanation of multi-head attention. We have revised the paper and added these details to Section 3.2.

2. **Dual-Stream Attention**: In response to concerns about dual-stream attention (Figure 2), we explained how the **attention interaction** between target and context representations helps prevent semantic entanglement, leading to improved target localization. We have revised the paper and added these details to Section 3.3.2.

3. **Training and Query Mechanisms**: We further clarified the **query mechanism** and its role in maintaining independent and interpretable target and context representations, highlighting how it enables adaptive fusion of features.

4. **Qualitative Results**: In response to the request for visual results, we added **visualizations** demonstrating the effectiveness of our method in challenging scenarios like motion blur and occlusion. We have revised the paper and added these details to Section 6.



### Reviewer i6rU

1. **Complexity and Interpretability**: We acknowledged the complexity introduced by design choices such as dual-stream attention, negative guidance, and initialization strategies. These were introduced to enhance model flexibility and long-term performance, especially in occlusion-prone environments.

2. **Hyperparameter Sensitivity**: We discussed the impact of training choices (like dense sampling and occlusion inclusion) on reproducibility and generalizability. However, we believe the framework's simplicity and adaptability make it easy to transfer to other datasets and domains.

3. **Failure Modes and Computational Trade-offs**: We provided further insights into **failure cases** (e.g., long-term occlusion) and discussed the **computational efficiency** of STARTrack, demonstrating how it outperforms other methods in terms of speed and resource consumption. We have revised the paper and added these details to Section 5.1.


### Reviewer LieF

1. **Novelty of STARTrack**: We clarified that while **token propagation** is not entirely new, STARTrack **eliminates the need for initial templates**, offering a unique approach to long-term tracking. This results in significant improvements in tracking performance.

2. **Performance Comparison**: In response to the reviewer’s concern about marginal gains, we introduced the **STARTrack-D** variant, which uses a more powerful backbone (DINOv2) and outperforms previous methods like SPMTrack. We have revised the paper and added these details to Section 5.2.

3. **Visualization of Tokens**: We added **visualizations** of the **target** and **context representation tokens** in action, providing clearer insights into how STARTrack effectively handles occlusions and other challenging conditions. We have revised the paper and added these details to Section 6.

### Reviewer cD5U

1. **Related Work**: We expanded the related work section to include **MeMOTR**, **STMTrack**, and **TCTrack**. We clarified that **PrDiMP** does not introduce temporal modeling as suggested. We have revised the paper and added these details to Section 2.1.

2. **Initialization Sensitivity**: We conducted an ablation study on target token initialization, showing minimal performance difference (~0.4%) across different enhancement strategies. We have revised the paper and added these details to Section 5.3.5.

3. **Computational Cost**: We clarified that the training cost of dense sampling is similar to sparse sampling, with no significant overhead.

4. **Hyperparameter Sensitivity**: We showed that STARTrack performs well across datasets with minimal hyperparameter adjustments, and longer sequence sampling improves performance.


---

Thank you for your valuable feedback. We have thoroughly revised and refined the paper based on your suggestions. We hope that these revisions address your concerns and improve the clarity of our contributions.

---

### Author Response · Authors · 2025-12-01
**Global response (1/2)**

## Contributions of the Paper

Our paper presents **STARTrack**, a novel framework for **spatio-temporal representation evolution** in **target-aware tracking**. The main contributions are:

1. **STARTrack Framework**: We propose a simple yet effective tracking model that utilizes a **token-based iterative propagation mechanism** to model the dynamic evolution of both target and context representations in videos. This avoids the sustained reliance on the initial template during long-term tracking, significantly improving performance in complex scenarios.

2. **Training Strategy**: A **novel training strategy** is introduced, incorporating the **Frame-wise Information Gain Principle (FIGP)** and **Dual-Stream Causal Representation Fusion (DCRF)**, which enhance the temporal consistency of representations. Additionally, for the first time, **occluded frames** are integrated into the training process, improving robustness under challenging conditions like occlusions.

3. **State-of-the-Art Performance**: Extensive experiments on benchmark datasets like LaSOT, GOT-10K, and TrackingNet demonstrate that STARTrack achieves state-of-the-art performance while maintaining a favorable balance between **accuracy** and **efficiency**.

---

### Meta-Review · Area_Chair_iDfP · 2025-12-25

**Summary:**

The paper proposes STARTrack, a video-level tracking framework employing iterative token propagation to model the evolution of target and context representations.

While the reviewers recognized the potential of the "evolutionary" paradigm and the extensive experiments, the consensus leans towards rejection. The primary reasons are the limited technical novelty (viewed as an incremental combination of existing ideas like token storage), the high complexity of the system (relying on numerous engineered training choices and hyperparameters), and presentation issues that hindered understanding for some reviewers. Although the authors introduced a stronger backbone variant (STARTrack-D) in the rebuttal to address performance concerns, the core methodological critiques regarding simplicity and novelty remain unresolved.

**Reviewer Concerns:**

**Addressed Concerns:**

1. Mathematical Notation (iYMN): The authors clarified that the notation in Eq. 2 follows standard Transformer formulations, addressing the specific confusion raised by Reviewer iYMN.

2. Missing Related Work (cD5U): Discussions on MeMOTR, STMTrack, and TCTrack were added.

3. Initialization Sensitivity (i6rU, cD5U): Ablation studies in the rebuttal (Tab. 1) demonstrated relative robustness to initialization strategies.

**Outstanding Concerns:**

1. Limited Novelty & Incremental Nature (LieF): Reviewer LieF noted that using dynamic tokens for temporal propagation is not novel (citing HIPTrack, SPMTrack) and viewed the dual-stream mechanism as an "incremental combination." The original performance gains were considered marginal (75.2% vs 74.9% for SPMTrack). While the authors added STARTrack-D in the rebuttal, this relies on a stronger backbone (DINOv2) rather than proving the superiority of the core tracking mechanism itself.

2. Complexity & Interpretability (i6rU): Reviewer i6rU highlighted that the method relies on "many engineered design choices" (two token types, negative guidance sign flips, specific initialization) and "specific training strategies" (dense sampling, occlusion inclusion, FIGP). This raises concerns about the conceptual interpretability and whether the performance stems from the method or the engineering tricks.

3. Presentation & Organization (iYMN): Despite the authors' appeal and clarifications, Reviewer iYMN's fundamental difficulty in following the logical flow and locating key mechanisms (like the dual-stream attention in Fig. 2) points to structural weaknesses in the manuscript's presentation that go beyond simple notation misunderstandings.

**Reviewer Scores:**

- Reviewer iYMN: 2 (Maintained). The reviewer's concerns about the "poor" organization and difficulty in understanding the architecture flow remain a significant barrier, despite the notation clarification.
- Reviewer LieF: 4 (Maintained). The core critique regarding the "incremental" nature of the contribution stands. The addition of a new backbone (STARTrack-D) in the rebuttal does not resolve the lack of novelty in the proposed mechanism itself.
- Reviewer i6rU: 6 (Maintained). While acknowledging the performance, the concerns about the "complexity" and "engineered training choices" prevent a strong endorsement.
- Reviewer cD5U: 6 (Maintained). The reviewer remains supportive but acknowledged the valid concerns regarding "insufficient depth" in related work and failure mode analysis.

---

### Decision · Program_Chairs · 2026-01-26

Reject